# SARS-CoV-2: Understanding the Transcriptional Regulation of ACE2 and TMPRSS2 and the Role of Single Nucleotide Polymorphism (SNP) at Codon 72 of p53 in the Innate Immune Response against Virus Infection

**DOI:** 10.3390/ijms22168660

**Published:** 2021-08-12

**Authors:** Niraj Lodhi, Rubi Singh, Satya Prakash Rajput, Quaiser Saquib

**Affiliations:** 1Clinical Research (Research and Development Division) miRNA Analytics LLC, Harlem Bio-Space, New York, NY 10027, USA; 2Department of Pharmacology, Weill Cornell Medicine, New York, NY 10065, USA; ruc2011@med.cornell.edu; 3School of Life Sciences, Warwick University, Coventry CV47AL, UK; Satya.Prakash@warwick.ac.uk; 4Department of Zoology, College of Sciences, King Saud University, Riyadh 12372, Saudi Arabia; quaiser.saquib0@gmail.com

**Keywords:** SARS-CoV-2, ACE2, TMPRSS2, p53, interferon, transcription, innate immune response, Single Nucleotide Polymorphism (SNP), codon 72 polymorphism of p53, TLRs (toll-like receptors), PARP1

## Abstract

Human ACE2 and the serine protease TMPRSS2 of novel SARS-CoV-2 are primary entry receptors in host cells. Expression of these genes at the transcriptional level has not been much discussed in detail. The ISRE elements of the ACE2 promoter are a binding site for the ISGF3 complex of the JAK/STAT signaling pathway. TMPRSS2, including IFNβ, STAT1, and STAT2, has the PARP1 binding site near to TSS either up or downstream promoter region. It is well documented that PARP1 regulates gene expression at the transcription level. Therefore, to curb virus infection, both promoting type I IFN signaling to boost innate immunity and prevention of virus entry by inhibiting PARP1, ACE2 or TMPRSS2 are safe options. Most importantly, our aim is to attract the attention of the global scientific community towards the codon 72 Single Nucleotide Polymorphism (SNP) of p53 and its underneath role in the innate immune response against SARS-CoV-2. Here, we discuss codon 72 SNP of human p53′s role in the different innate immune response to restrict virus-mediated mortality rate only in specific parts of the world. In addition, we discuss potential targets and emerging therapies using bioengineered bacteriophage, anti-sense, or CRISPR strategies.

## 1. Introduction

In December 2019, a coronavirus disease (later named COVID19) was reported in Wuhan in China. In the beginning, it was thought to be endemic and restricted only to that area, but later, it spread rapidly throughout China and then to many other parts of the world [1,2]. The regulatory body, the World Health Organization (WHO), declared COVID19 a pandemic on 11 March 2020, and as of 3 June 2021, there have been 171,292,827 confirmed cases from 216 countries or areas, with more than 3,687,589 confirmed deaths with a recovery rate around 89% (WHO, 2020). The United States of America (USA) is among the worst affected by the pandemic; according to the Centers for Disease Control and Prevention (CDC) and the John Hopkins University and Medicine (Corona Virus Resource Center), as of 20 March 2021, more than 29,782,754 total cases have tested positive for the Severe Acute Respiratory Syndrome Coronavirus-2 (SARS-CoV-2) and more than 541,913 have died, which is around one-fourth of global deaths due to CoVID19. The severity of the infection rate can be understood by the fact that on average 71,000 new cases are diagnosed as positive cases a day (a decline from more than 100,000 cases per day). Two-thirds of the 230,000 new cases that were reported worldwide on 16 July 2020, came from just four countries—the United States, Brazil, India, and South Africa. At the beginning of June last year, the states New York, Connecticut, and New Jersey collectively accounted for about 30% of total cases and 42% of deaths of the nation. This widespread sustained coast-to-coast transmission of SARS-CoV-2 within the United States highlights the critical need for local surveillance [3].

Virus entry in the host cells depends on the spike (S) protein of coronaviruses. The S1 subunit of the S protein binds to surface receptor of host cells, which helps virus attachment to the surface of host cells. SARS-CoV-2 uses the angiotensin-converting enzyme 2 (ACE2) surface receptor of host cells for entry into host cells similar to other viruses of its family [4]. After receptor attachment, SARS-CoV-2 S is processed by a plasma membrane-associated type II transmembrane serine protease, TMPRSS2, for S protein priming (S protein cleavage at the S1/S2 and the S2′ site required for fusion of viral and cellular membranes, a process controlled by the S2 subunit) [5,6,7]. Understanding ACE2 and TMPRSS2 protein expression at the transcription level in the human cells could reveal important insights into differential susceptibility to influenza and coronavirus infections and novel targets for therapeutic purposes. The study of Stopsack et al. [8] suggests that TMPRSS2 protein is more heavily expressed in the lungs’ bronchial epithelial cells than in surfactant producing type II alveolar cells and alveolar macrophages. Therefore, their immune response responds according to virus entry in host cells. Regulatory elements (ISRE) and STAT1/2 protein binding elements in the promoter of ACE2 control its expression by binding to the ISGF3 complex, and indeed, ACE2 is upregulated after exposure to Type I IFN [9]. 

After entering the virus or bacteria in cells, the innate immune response of the host cell responds to invader cells. Extracellular Toll-Like Receptors (TLRs) recognize the intracellular contents of bacterial invaders damage-associated molecular patterns (DAMPs), while downstream macrophages polarize to the classical pathway and facilitate the pro-inflammatory response (discussed later). Conversely, intracellular TLRs recognize viral RNA or oxidized phospholipids from invading viruses. Once SARS-CoV-2 enters through ACE2 and TMPRSS, internal TLRs initiate the type I interferon and pro-inflammatory pathway and evoke the antiviral state in host cells to clear the virus. However, respiratory viruses cleverly disguise to block the type I IFN activation [10,11]. This signal transduction by autocrine or paracrine drives the production of cytokines, which activates antiviral gene expression in self or neighboring cells to promote macrophage polarization and reactive oxygen species (ROS) in cells. In addition to intracellular TLRs, one study reported the role of extracellular TLR4 signaling and oxidative stress subsequently ROS production is a key pathway of Acute Lung Injury (ALI) after SARS or H5N1 avian flu virus infection [12].

Macrophages contribute to both viral control and tissue damage as well as DNA damage of virus and host cells. Type I IFN pathway promotes intracellular antiviral defenses in neighboring cells, which limits viral dissemination, releases IL6 and IL1 β, and recruit neutrophils and cytotoxic T cells [13]. The interferon (IFN) pathway induction of interferon-stimulated genes (ISGs) is essential for host antiviral defense in mice and humans [14,15,16,17,18]. There are three distinct types of IFNs: Type I IFNs (IFNα and IFNβ), type II IFNs (IFNγ), and type III IFNs (IFNl) [19]. All three IFNs pathways involve the hetero- or homodimers of STAT1/STAT2 and IRF proteins and JAK/STAT signaling to induce pro-inflammation; however, each pathway has a non-redundant role to provide host defense [19,20]. By dysregulating the type I IFN pathway, Inflammatory Monocyte-Macrophage (IMMs) responses cause lethal pneumonia in SARS-CoV-infected mice [21]. By ablating functions of STAT proteins in mice, Alternatively Activated Macrophages Enhances Pathogenesis (AAMEP) induces during SARS-CoV infection [22]. This signal transduction drives the production of cytokines ‘storms’, activates antiviral gene expression programs in neighboring cells, and recruits additional innate and adaptive immune cells with diverse roles in antiviral immunity, tissue repair, and homeostasis. 

Overall, our review provides unique information to better understand the transcriptional expression of ACE2 and TMPRSS2, and different immune responses or mortality rates in people of different parts of the world. We describe and discuss the pathways that play a major role of type I IFN and JAK/STAT pathways in the regulation of ACE2 and possible DNA damage-dependent PARP1-regulated TMPRSS2 gene expression. Importantly, we focus on the possible role of Single Nucleotide Polymorphism (SNP) at codon 72 of p53 in the different inflammatory response or mortality rates in different parts of the world. The dysregulated immune response of COVID19 patients pose an enigma to the global scientific community [23,24]. Yet, there is no satisfactory answer to why these different immune responses as well as death rates among the COVID19 patients are discriminated all over the world. We understand that SNP at codon 72 at p53 polymorphism holds answers to the riddle and has a role in the broad spectrum of immune response in patients. It is high time to explore further on underneath role of SNP at codon 72 of p53 polymorphism in the current pandemic. In the end, we describe new potential targets and bioengineered ribozyme and a riboswitch-based antisense or bacteriophage-based vaccine for SARS-CoV-2.

## 2. Transcriptional Regulation of ACE2 by ISGF3 Assembly Complex in Type I IFNs and the JAK/STAT Pathway

A conserved DNA sequence element was reported in type I ISG promoters and characterized as the IFN-stimulated response element (ISRE) [25]. It is binding site of Interferon Stimulated Gene Factor 3 (ISGF3) complex, once it translocated in the nucleus after IFN pathway stimulation [26]. The ISRE consensus DNA sequence, 5′-AGTTTCNNTT TCNC/T-3′, is found at the promoters of most direct type I/III ISGs. In the cytoplasm, the pre-assembled ISGF3 complex translocated to the nucleus and binds to ISRE specifically through its binding partner’s interferon regulatory factor 9 (IRF9), STAT1, and STAT2. It was reported earlier by [27,28] that the regulatory element in ACE2 promoter is located at the upstream region (-516/-481) to the transcription start site (TSS). The sequence of this element is ATTTGGA, homologous to ISRE consensus; however, Kuan et al. [28] reported that it was not responsible for pro-inflammatory cytokines TNF-α or TGF-β11, although the authors did not check the stimulation through the TLR4 receptor. In recent years, ISRE variants (5′-CTTTNNCTCT-3′ or 5′-CTCTNNCTCT-3′) have also been identified by several groups [29], which suggests that it may be indeed the binding site for the ISGF3 complex in the ACE2 promoter [29].

Using the www.interferome.org online database, we have generated transcription factor binding regions spanning from −1500 to +500 bp from the transcription start site of ACE2 and found STAT1, IRF1, and IRF9 sites in the proximal region of the promoter (Figure 1A). It shows there are binding DNA elements from the ISGF3 complex in the ACE2 promoter. In addition, Gene Ontology (GO) analysis shows, majorly involves in the viral entry into the host cell, response to virus infection, virion attachment, and binding of a host cell receptor, regulation of inflammatory response, and cytokine production (Table 1). The Venn diagram shows the number of genes regulated by one or more IFN type (Type I or II) (Figure 1B). There are several studies or datasets are available publicly for type I compared to type II or III; therefore, to avoid biasness results of new studies of SARS-COV-2 should be interpreted carefully and consider very low or negative gene expression results [30].

Recent reports show elevated interferon pathways in the lung cells of COVID-19 patients [2,31]. To evaluate the cytokine and interferon response to SARS-CoV-2 infected AAV-ACE2 mice, Israelow et al. [32] performed RNA sequencing from infected lung at 2 days post-infection (DPI). The results of their study suggest infected mice have increased expression of cytokines and ISGs, and in fact, the majority of the differentially expressed upregulated genes were either ISGs or cytokines. However, Barker and Parkkila [33] reported that several transcription factors bind in the proximal promoter (400 bp upstream to TSS), including CDX2, HNF1A, FOXA1, SOX4, TP63, HNF4A, DUX4, FOXA2, NR2F6, and SOX11 for tissue-specific expression, but surprisingly, none of the factors are directly related to type I IFNs or JAK/STAT pathway, although they characterized and determined cell-specific expression and functions very extensively.

Altogether, transcription initiation of ACE2 takes place after binding of the ISGF3 complex, and formation of the complex includes IFNs binding to its receptors to induce JAK/STAT signaling. Therefore, viral infection encourages the positive regulation of ACE2 transcription.

**Figure 1 ijms-22-08660-f001:**
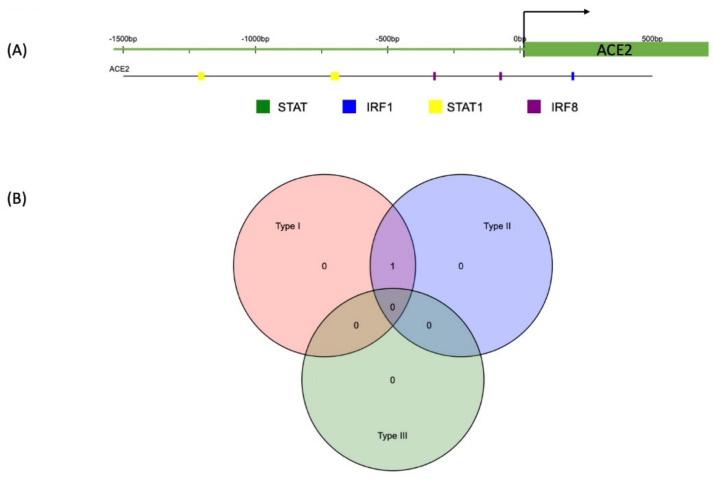
Transcription binding sites in the promoter of ACE2 and overlapping of type I and type II IFNs pathways in the regulation of ACE2. (**A**). Using www.interferome.org, we generated transcription binding sites in promoter region of ACE2. The location of the transcription factors on each of the genes from the region spanning −1500 bp to + 500 bp from the start site. Each coloured box represents a specific transcription factor, the key for which is provided below the graphic. TRANSFAC* predicted a match between the transcription factor and its predicted binding site [34]. (**B**). The Venn diagram shows the number of genes regulated by one or more IFN type (Type I, II or III). It should be noted that there are far more datasets for genes regulated by type I than for types II or III; this imbalance introduces the risk of false negatives and bias for the under-represented types II and III. Therefore, caution is encouraged in interpreting low or negative results from these types.

**Table 1 ijms-22-08660-t001:** Gene Ontology (GO) Analysis (Biological Process) of ACE2.

Accession	Term Name	Term Definition	Gene Count	*p* Value *
GO:0006508	Proteolysis	The hydrolysis of proteins into smaller polypeptides and/or amino acids by cleavage of their peptide bonds. [GOC:bf, GOC:mah]	1	0.0106
GO:0046718	Viral entry into host cell	The process that occurs after viral attachment by which a virion, or viral nucleic acid, breaches the plasma membrane or cell envelope and enters the host cell. The process ends when the viral nucleic acid is released into the host cell cytoplasm. [GOC:jl, PMID:12142475]	1	1.3 × 10^−4^
GO:0046813	Virion attachment, binding of host cell surface receptor	The process during virion attachment where a virion binds to a host cell receptor, resulting in a conformational change of the virus protein. [ISBN:0879694971]	1	7.02 × 10^−5^
GO:0042127	Regulation of cell proliferation	Any process that modulates the frequency, rate, or extent of cell proliferation. [GOC:jl]	1	0.00281
GO:0019229	Regulation of vasoconstriction	Any process that modulates the frequency, rate, or extent of reductions in the diameter of blood vessels. [GOC:jl]	1	2.81 × 10^−4^
GO:2000379	Positive regulation of reactive oxygen species metabolic process	Any process that activates or increases the frequency, rate, or extent of reactive oxygen species metabolic process. [GOC:mah]	1	4.21 × 10^−4^
GO:0009615	Response to virus	Any process that results in a change in state or activity of a cell or an organism (in terms of movement, secretion, enzyme production, gene expression, etc.) as a result of a stimulus from a virus. [GOC:hb]	1	0.00302
GO:0050727	Regulation of inflammatory response	Any process that modulates the frequency, rate, or extent of the inflammatory response and the immediate defensive reaction (by vertebrate tissue) to infection or injury caused by chemical or physical agents. [GOC:ai]	1	6.11 × 10^−4^
GO:0003051	Angiotensin-mediated drinking behavior	The drinking behavior that is mediated by the action of angiotensin in the brain. Angiotensin stimulates the brain centers that control thirst. [GOC:mtg_cardio]	1	4.01 × 10^−5^
GO:0032800	Receptor biosynthetic process	The chemical reactions and pathways resulting in the formation of a receptor molecule, a macromolecule in combination with a hormone or neurotransmitter, drug, or intracellular messenger to initiate a change in cell function. [GOC:mah]	1	1.2 × 10^−4^
GO:0001817	Regulation of cytokine production	Any process that modulates the frequency, rate, or extent of production of a cytokine. [GOC:add, ISBN:0781735149]	1	1.5 × 10^−4^
GO:0002005	Angiotensin catabolic process in blood	The chemical reactions and pathways resulting in the breakdown of angiotensin in the blood. [ISBN:0721643949]	1	3.01 × 10^−5^
GO:0003081	Regulation of systemic arterial blood pressure by renin-angiotensin	The process in which renin-angiotensin modulates the force with which blood passes through the circulatory system. [GOC:mtg_cardio]	1	7.02 × 10^−5^
GO:0042312	Regulation of vasodilation	Any process that modulates the frequency, rate, or extent of increases in the diameter of blood vessels. [GOC:jl]	1	9.02 × 10^−5^

Using www.interferome.org, we generated functional pathways of ACE2. The enrichment of each gene ontology term was tested for significance using the hypergeometric mean. The *p* value returned represents the probability of that each term would be the observed with the given frequency, had the genes been drawn at random from across the entire genome. A low *p* value * (less than 0.05) for an ontological term indicates a low probability that so many hits to that term would have been observed if the results had been due to random effects, and such terms might be considered to be enriched in the results [35,36].

## 3. Transcriptional Regulation of TMPRSS2 by PARP1 Binding

Our previous study showed [37] that PARP1 binds to genes that play a regulatory role in the type I IFN signaling pathway (Figure 2A–D). PARP1 binds either near to TSS (up or downstream) of promoter of the gene. More importantly, our ChIP-seq data also show that interferon signaling genes have PARP1 binding during the mitosis of the cell cycle. PARP1 binds in mitosis as an epigenetic factor to those genes whose expression important for the extracellular region, membrane integration, and cell surface, or in extracellular space functions during the exit of mitosis and activates its transcription through auto-poly(ADP-ribosyl)ation. 

PARP1 binds to the upstream region of TMPRSS2′s promoter during mitosis (Figure 2A), as an epigenetic factor PARP1 memorizes the cell for its early-stage expression of genes when cell exits from mitosis [37], similar results have been shown by other groups in different genes [38,39,40]. It is possible that PARP1 reactivates the transcription of TMPRSS2 during viral infection, however further research is needed to make any statement. PARP1 also binds to type I IFNβ, STAT1, and STAT2 either in up- or downstream regions of these genes (Figure 2B–D) [5,41,42,43]. PARP1 binding to Drosophila histone variant H2Av, near transcription start site (TSS) is reported to control the initiation of transcription by remodeling of the nucleosome at the core promoter of the HSP70 gene [44]. PARP1 binds at 182 bp downstream to TSS of IFNβ, upstream to TSS (−279 bp) of STAT1, and at −1566 bp upstream to TSS of STAT2. Further research is needed to reveal the role of PARP1 in the transcription initiation and expression of these genes. It has been reported that transcriptional activation of the IFNβ in response to virus infection requires the assembly of an enhanceosome, which recruits the transcription factors and chromatin remodelers to slide the nucleosome over the core promoter region to a downstream position, required for transcriptional activation [45]. It is possible that PARP1 colocalizes with the nucleosome of IFNβ and helps its sliding to activate transcription. It has been reported that PARP1 colocalizes with Drosophila histone variant H2Av nucleosome on the promoter of HSP70 gene in Drosophila [44] and JIL-1 kinase mediated changes in H2Av nucleosome conformation cause chromatin de-condensation by PARP1 enzymatic activity [46].

Using the www.interferome.org online database, we have generated transcription factor binding regions spanning from −1500 to +500 bp from the transcription start site of human TMPRSS2 and found the IRF1 binding site (Figure 2E). It is possible that IRF1 binding in the upstream region of TMPRSS2 plays a pivotal role in the induction of this gene transcription during viral infection. However, further research is needed to draw conclusions.

In the steady-state conditions, PARP1 proteins are associated with chromatin and are accumulated in nucleoli [47,48,49,50,51,52]. PARP1 can function as the transcription repressor; as reported by Lodhi et al. 2014 [37], genes have a physical interaction with PARP1 in the interphase of the cell or activator, depending on its enzymatic activity [51]. On the one hand, enzymatically silent PARP1 binds to the nucleosomes to increase the chromatin condensation for transcription inhibition [51], but on the other hand, histone variants containing nucleosome can stimulate PARP1 enzymatic activity in chromatin, leading to chromatin de-condensation for transcription activation [44]. Activation of chromatin-associated PARP1 leads to the assembly of poly(ADP-ribose) (pADPr) via transfer of ADP-ribose to various substrates, including existing pADPr chain, PARP1 itself, and other acceptor proteins. It is characterized by auto-poly(ADP-ribosyl)ation for PARP1 and poly(ADP-ribosyl)ation for others. Accumulation of negatively charged moieties of pADPr on the pre-existing pADPr chain or glutamate residue of acceptor proteins neutralizes the positively charged groups which are bind to DNA (protein-DNA complex), it causes repulsion from negatively charged DNA, eventually it mediates chromatin de-condensation, and stimulates transcription [44,46,47,53,54]. 

Transcription stimulation of TMPRSS2 through enhanced poly(ADP-ribosyl)ation needs to be explored experimentally. Although it makes sense, virus infection leads to the exponential expression of pro-inflammatory cytokines, and iNOS subsequently produces nitric oxide (NO) [55]. There are several reports which have shown that viruses increase iNOS through the subsequent activation of TLR3 and NO [56]. Activation of iNOS and NO production inhibit viral replication (HIV-1, coxsackievirus, influenza A and B, and rhino virus) [57]. Nitric oxide and iNOS are potential sources of production of ROS [58,59,60,61]. These species trigger the eradication of viral or bacterial infections on the one hand and modulate immunosuppression during tissue-restoration and wound-healing processes on the other. ROS-mediated DNA damage triggers activation of DNA repair machinery, including PARP1 [62]. Activated auto-poly(ADP-ribosyl)ated PARP1 moves away from the DNA binding site and it may increase the transcription of TMPRSS2, which possibly facilitates further virus entry in cells (Figure 2F). Therefore, it is possible to manipulate transcription expression by compromising PARP1 binding to TMPRSS2 using PARP1 inhibitors. It has been shown that ROS could activate NFκB, and longer exposure to ROS inhibits NFκB phosphorylation and activation in T cells [63,64].

## 4. Modulation of the Immune Response to a Virus

### 4.1. Immunogenic Response Training to Develop Immunity, Reduce Susceptibility and the Severity of SARS-CoV-2 Infection

Bacillus Calmette-Guerin (BCG) vaccination is mandatory in developing countries (Southeast Asian countries) to prevent infection of *Mycobacterium tuberculosis*, which is very common there. It is well established that vaccination of attenuated tuberculosis bacteria boosts the innate immune response not only for mycobacteria but non-specifically for other viruses or bacteria. In the current pandemic, the low infection rate and deaths in these countries is likely because of their immune response, which fights more strongly to clear viruses from the cell in comparison to European countries and the United States. Therefore, BCG vaccination protects consistently from subsequent infections [65]. In a small study in Indonesia, multiple doses of BCG vaccination in elderly people exhibited a significant reduction in the number of respiratory tract virus infections as compared with a placebo [66]. Another clinical trial in Japan found that BCG vaccination in people makes them less susceptible to pneumonia [67].

In the current pandemic, these results seem to prove that BCG vaccination protects against viral pathogens, including respiratory syncytial virus, human papilloma virus, and herpes simplex virus [68]. Studies in mice show that BCG protects non-specifically against influenza virus infection and human studies show it protects against influenza pneumonia by reducing inflammation [69,70]. Last year, in first wave of pandemic it was clear that death rates of CoVID19 patients were much less in BCG vaccinated countries than UK, European and North American countries.

### 4.2. Gaining Immunity through Genetic Variation during Evolution

Genetic polymorphisms are responsible for population diversity and SNPs have significant biological consequences by predicting diseases such as breast cancer or acute lymphocytic leukemia (ALL) [71,72,73]. As the transcription factor human tumor suppressor gene p53 plays a major role in maintaining cellular integrity by activation of the apoptosis pathway and cell cycle arrest in response to DNA damage or other cellular insults. As the tumor suppressor, its role in cancer development is extensively studied; in most cancers, mutation in p53 prevents its functions, it is well documented [74]. NFκB and STATs regulate transcription induction of the p53 gene during viral infection [75,76].

Its cellular function also includes orchestrating the cell response to a broad range of stresses including immune response to bacterial or viral infections, oncogenic, genotoxic, and nutritional stresses. As a nuclear protein and transcription factor, p53 is evolutionarily conserved across Drosophila to human and plant and activates or downregulates the transcription of target genes that are responsible for many cellular responses against different cell insults. In human populations, several studies reported that SNP at codon 72 of p53 affects cell response to nutritional deprivation, immune response, and mice ageing [77,78,79]. It is well known that the effect of change in nucleotide at second position of codon 72 at exon 4 (rs1042522—CCC to CGC) resulting in a substitution of proline (Pro, P) to arginine (Arg, R) in the P-rich region domain of p53 [80]. Interestingly, the arginine 72 variant (R72) and proline 72 variant (P72) allele frequencies vary in different ethnic populations and geographic regions of earth at the equator or latitudinally. The P72 allele is more frequent in the human population near the equator, whereas R72 is more frequent in a linear manner with latitude (Figure 4A). In an estimate, about 64% of African-Americans and 58% of Hispanic-Americans express the P72 variant. Conversely, only 32% of White Americans have the P72 variant, in which 10% are homozygous to this allele. The human population that lives near the equator is about 40% homozygous for P72, and this percentage drops higher up longitudinally and R72 frequency increases in people [81]. Of the 13 countries that lie on the equator, 7 are in Africa—the most of any continent—and South America is home to 3 of the nations. The remaining countries are island nations in the Indian and Pacific oceans. Countries intersected by the equator experience much warmer temperatures and are more humid year-round than the rest of the world, making them vulnerable to many infectious diseases including influenza, cholera, malaria, tuberculosis, and respiratory diseases.

From ancient times, p53 carried the P72 variant and it is believed that the R72 variant arose around 30,000 to 50,000 years ago; however, it remains to be answered why this variant is included in the human genome [82]. The results of several studies indicate as one possibility that the P72 allele is selected for near the equator because the immune challenge is greater there compared to far-north countries, as well as innate immune response associated with the P72 allele. In support of this notion, the P72 variant is generally associated with longevity, even following noncancerous illness [83,84]. 

It largely remains elusive why the population of the R72 variant is frequent in northern latitudes; recent publications showed that the R72 variant is metabolically tolerated in the northern population, where nutrients are efficient (discussed later). A study in knock-in humanized p53 mice, encoding either the P72 or R72 variants, showed that significantly altered levels of NFκB-dependent apoptosis in different tissues of P72 or R72 mice [85]. Furthermore, P72 mice have a significantly enhanced response to inflammatory challenge compared to that of R72 mice. In another study, by looking at the different gene transcriptional responses between the two variants, it shows that P72 and R72 respond dramatically differently to innate immunity stress. This difference is regulated by a subset of genes known to control the inflammatory response, including some of the NFκB target genes, which are better activated by the P72 variant [86]. 

“If P72 has a better response with innate immunity, then it might make sense why this variant is selected for near the equator during evolution, where malaria and other infections are more common, since the P72 variant is more common in African-Americans and Asians. This may help to explain ethnic disease disparities”, stated Prof. M. Murphy (WISTAR Institute, Philadelphia) at National Institute of Environmental Health Sciences (NIEHS) Laboratory of Molecular Genetics (LMG) Fellows Invited Lecture Series on 7 May 2012. 

R72 polymorphism confers increased cell survival in response to nutrient deprivation, it holds true for the northern hemisphere where people are deprived of nutrients because of scarcity, it shows biased selection of R72 of nature [87]. Furthermore, another study showed R72 polymorphism mice gain weight on a high-fat diet (HFD); additionally, increased obesity coincided with increase glucose intolerance and insulin resistance, which subsequently leads to increase susceptibility to type 2 diabetes [88]. The study also suggests that R72 mice tolerate a diet rich in fat and sugar better, while P72 mice tend to develop diabetes on this regimen, which may explain the higher risk for diabetes in African-American populations. Furthermore, it suggests that, during evolution, there may have been selection for the P72 allele in environments with more ultraviolet light or warmer winter temperatures.

In cancer research, P72 mice escaped from tumor development, had a longer lifespan than R72 mice, and showed a delay of age-associated phenotypes. In a recent large study on mice, on the ageing process, it was concluded that P72 mice have a better ability to retain the self-renewal of stem or progenitor cells compared with R72 mice [89]. It is well aligned with the other study of human populations with the P72 allele have a longer lifespan [90]. People with R72 are more susceptible to tuberous sclerosis complex (TSC) tumorigenesis, angiomyolipomas and lymphangioleiomyomatosis (LAM) diseases [91]. In fact, 90.3% patients of these diseases have R72 variants and plays major role in TSC tumorigenesis through enhance transactivation of NOTCH1 and NODAL pathways in mouse model of human R72 (88). 

A similar pandemic study, reported in 2012 [92], about a novel swine-origin A/H1N1 influenza virus, was identified in Mexico and the United States in late April 2009. There were more than 35,000 cases reported from 70 countries around the world and nearly 200 confirmed deaths [92]. Despite the highest infectious disease burden and the lack of a proper health care system, temperate and tropical regions (equatorial regions), where nearly half of the world population lives, were less affected by the 2009 influenza pandemic.

Taken together, the development of type 2 diabetes, obesity, diseases associated with inflammation, or heart-related diseases in the R72 population makes them more susceptible to SARS-CoV-2 infection and morbidities. Therefore, we see a difference in mortality rate, increasing at higher latitude and decreasing when at or near the equator. 

## 5. Modulation of Immune Response Genes to Develop Therapy

In order to get acquired immunity, innate immunity activation plays a major role. Inflammation alerts and protects cells from subsequent infections and restricts further spreading to neighboring cells. It is a coordinated complex process of the pathways that eventually induce other cellular functions to repair the damages of cells. One of them, type I IFN pathway is crucial to the innate immune response against invader pathogens [93]. 

In the innate immune response, Pattern Recognition Receptors (PRRs) are the first responders and bind to various PAMPs and DAMPs. TLRs are among the most important component of the PRRs family. TLRs recognize the invaders PAMPs and DAMPs based on their locations in the cell and divided into two cell surface and intracellular TLRs. TLRs present on the cell surface is TLR1, -2, -4, -5, -6, and -10. Intracellular TLRs include TLR3, -7, -8, -9, -11, -12, and -13, and are present at endosomes or other cellular organs [94,95]. Cell surface, TLRs recognize lipoproteins, lipids, and proteins of bacterial or microbial membranes. One of the most studied, TLR4, which is a major part of the innate immune response and recognizes bacterial lipopolysaccharide (LPS) or endogenous molecules from injured and necrotic cells (DAMPs), and activates the downstream proinflammatory pathway [96]. Other TLRs, i.e., TLR1, -2 or -6, recognize various PAMPs, including lipoteichoic acids, lipoproteins, and peptidoglycans [94].

To begin the downstream innate immune pathway, intracellular TLRs recognize nucleic acids derived from viral and bacterial infections or self-nucleic acids in case of auto-immunity diseases [97]. TLRs at endosomes including TLR3, -7, -8, and -9 recognize double-stranded RNA (dsRNA), single-stranded RNA (ssRNA), and dsDNA of viral PAMPs [98,99]. TLR7 recognizes ssRNA and begin the proinflammatory pathway to produces type I IFNs and pro-inflammatory cytokines [100,101,102] and TLR8 responds to viral and bacterial RNA and activate the proinflammatory pathway through interferon regulatory factor 3 (IRF3) and IRF7 [103,104]. TLR9 recognizes unmethylated CpG bacterial or viral DNA [105]. Phylogenetically TLR7, -8, and -9 are close to each other and with TLR3, they are located in endosomal compartments exposing the TIR domain toward the cytoplasm. TLRs require ligand internalization to the endosome and activation of the intracellular signaling pathway of type I IFNs through IRFs [106]. In antigen-presenting cells (APCs), viral PAMPs activate the pro-inflammatory pathway and it is important that endosomal TLRs are restricted to activate the type I pathway only [107]. 

Nearly all TLRs firstly recruit a common adapter protein MyD88, considered as a fundamental recruiter protein to activate the innate immune response [108]. It forms the platform to recruit further proteins, including IL-1R-associated protein kinases (IRAK) 1, 2, 4, and M, TAB2, and TNF receptor-associated factor 6 (TRAF6), which finally leads to nuclear translocation of the pro-inflammatory transcription factor, nuclear factor kappa-B (NFκB) [109], activator protein 1 (AP-1) [110], and IRF3 [111]. Each protein of this complex is responsible to activate the transcription of specific pro-inflammatory cytokines tumor necrosis factor-alpha (TNF-α), interleukin (IL)-1β and IL-6, and type 1 IFNα and β [112].

By autocrine or paracrine signaling, Type I interferons bind to its receptors of the self-cell or neighboring cell and activate a JAK/STAT signaling to form the ISGF3 complex. By receptor engagement, JAK1 and TYK2 phosphorylate intracellular domains of IFNα/β receptors, which provide recruitment sites for the unphosphorylated SH2 domains of STAT1 or STAT2 [113,114,115]. These phosphorylated STAT1 or STAT2 to form the homo- or heterodimer ISGF3 complex with IRFs. 

This pre-assembled complex translocated to the nucleus and binds to a consensus ISRE of type I ISG promoters. As discussed earlier, the ISRE consensus DNA sequence, 5′-AGTTTCNNTT TCNC/T-3′, is found in promoters of type I ISGs, including the iNOS gene and IL2 and IL27, and induces exponentially to polarize macrophages in the classical pathway (M1). These M1-polarized macrophages are released out of cells to phagocytose the invader cells, such as viruses or bacteria, and release DAMPs and bind to TLR4 receptors, which further enhance the MyD88-dependent pro-inflammatory pathway to induce type I IFNs. Consensus ISRE in the promoter of ACE2 binds to the ISGF3 complex and initiates the transcription and functions as the receptor for SARS-CoV-2. After internalization of the virus in cells via endosomes, TLR9 receptors recognize viral PAMPs and trigger the pro-inflammatory pathway by acting as a platform to recruit the IKKS family and TRAF6 proteins, with subsequent activation of NFκB to bind its element in the promoter of type I interferons.

The type I IFN pathway plays a central role against virus infection by the host innate immune response, which leads to increased expression of ISGs and cytokine production to promote proinflammatory response against viruses [116,117,118]. Simultaneously, TMPRSS2 expression is initiated by auto-poly(ADP-ribosyl)ation of PARP1 (Figure 2F). This enzymatic activation of PARP1 is caused by ssDNA damage by induction of reactive oxygen species (ROS) in viral/bacterial infected cells. ROS is induced by exponentially expression of iNOS and production of nitric oxide (NO). Auto-poly(ADP-ribosyl)ated PARP1 positively controls the TMPRSS2 expression and facilitates viral entry into target cells by viral S protein priming.

## 6. Prevention and New Targets to Develop Therapy

Although it is reported SARS-CoV-2 cell entry is blocked by clinically proven inhibitors of ACE2 and TMPRSS2 [43], but identification of new targets is very necessary to curb the infection or mortality caused by SARS-CoV-2. Below, first, we discuss the preventive measures based on evidences or by our understanding of the geographical distribution of p53 polymorphism in the human population.

### 6.1. Prevention 

#### 6.1.1. Vaccination

Since ancient times, vaccination has been an effective tool to boost the host’s innate immunity, vaccines protect from not only from specific but also from non-specific infections for a long time. With subsequent infection, these cells respond faster to induce an anti-viral or bacterial state in neighboring cells to reduce the further transmission and severity of infection (Figure 3).

Vaccination is a type of training the immune system to respond robustly against invaders, which is very useful during the current pandemic to protect from SARS-CoV-2 infections and severity [119,120]. BCG vaccination is proved to be effective to reduce the spread of infections even for a limited period, and restriction of the further infections was very important in the beginning of the pandemic. Among the other approaches, such as complete lockdown, social distancing, and face masks, it can contribute significantly to prevent fast spreading in addition to trained immune response [121]. India is the best example, where mandatory BCG vaccination, complete lockdown, a ban on air travels, and social distancing implemented at the beginning of the year contributed significantly to save human lives compare to Western countries.

It is a fact that the percentage recovery rate of patients is higher in South-East Asian countries, where BCG vaccination is mandatory to boost the immune system against *M. tuberculosis* infection.

#### 6.1.2. Codon 72 Polymorphism of p53

Based on published research papers [76,77,78,79], the R72 population is more susceptible or vulnerable to viral infection, because their pro-inflammatory immune response is not so strong as the P72 population (Figure 4A). Therefore, the R72 population should comply with preventive measures, such as social distancing, use of masks, and no touching of the mouth and nose. It can be checked by the severity of infection rate and mortality in different population. Again, countries near the equator, such as African and Southeast Asian countries, have a much lower mortality rate compared to European and American countries (Figure 4B). This can be easily seen when looking at the effect SARS-CoV-2 has had in Kenya: the first wave of the epidemic passed its peak with just over 600 mortalities, much fewer than expected. It left the scientific community of the world baffled. The Kenya Medical Research Institute and Wellcome Trust Research Program suggested in their model that 30–40% of the population of the Kenyan cities Nairobi and Mombasa have been exposed (Figure 4B).

If we compare the same proportion of people in the hardest-hit parts of New York City, not only is the death rate relatively low, but in addition, about 90% of Kenyans who have tested positive were asymptomatic. “If you do compare with what has happened elsewhere, then, yes, Kenya has dodged a bullet and nobody knows without vaccine and medicines; how did this happen?” Dr. John Ojal of Kenya Medical Research Institute said in a statement, issued by his institute. 

Through our interpretation, as well as other’s research findings, we understand that this is because the equator line passes through the central part of Kenya. Hence, it can be hypothesized that more of the population has the P72 allele there, and thus, they have a better innate immune response against SARS-CoV-2, as we discussed earlier. 

### 6.2. New Targets

#### 6.2.1. PARP1

During mitosis cell division, PARP1 bookmarks genes which are essential to express early in newly divide daughter cells [33]. These genes are responsible for establishing cell niche, adhesion, communication, and proliferation, collectively termed as cell identity genes. In addition, PARP1 bookmarks pro-inflammatory and apoptosis and tumorigenic genes, including proto-oncogenes and tumor suppressor genes (Figure 2A–E) [37,38].

Blocking the enzymatic functions of TMPRSS2, IFNβ, and STAT1/2 by their expression at transcription level using inhibitors could be a novel and better strategy. However, the current PARP1 inhibitor has been approved by Food and Drug Administration (FDA) for ovarian and breast cancer has as a limitation that it has to compete with nicotinamide adenine dinucleotide (NAD^+^), which leads to a non-specific effect on other NAD^+^-dependent enzymatic activities, such as Sirtuins functions on histone proteins de-acetylation. To circumvent this, Professor Alexei Tulin’s lab at Fox Chase Cancer Centre, Philadelphia identified structurally new non-NAD^+^-like inhibitors that block PARP1 activity in cancer cells with greater efficacy and potency than classical PARP1 inhibitor currently used in the clinical practices [44]. The efficacy of these new inhibitors is better than NAD^+^-dependent inhibitors in prostate cancer [122] and breast cancer [123].

In the current pandemic, there is no standard drug to treat COVID-19; however, several groups are trying previously established drugs to treat other inflammatory diseases, and most of the time they have found better outcomes (Dexamethasome) [124]. India’s premier institute, i.e., the Indian Council of Medical Research (ICMR), and the Department of Biotechnology (DBT) jointly started human trials of the vaccine, but there are no pre-clinical data available. In the past few months, the second wave of SARS-CoV-2 in India has resulted in a severe health crisis. India has been listed among the worst-hit countries in this pandemic and ranks third in the world in both total infections and the number of new infections recorded each day after the USA and Brazil. It is presumed that there is no harm to use PARP1 inhibitors on COVID-19 patients, though we may we find better outcomes as well.

#### 6.2.2. TLR3, -4, and -9

TLR3, -4, and -9 are not new targets, but still remains inconclusive because of the lack of inclusion of a large number of patients in previous clinical trial studies of COVID-19. For example, hydroxyquinoline (HCQ) is a TLR9 inhibitor used as a clinically proven treatment for malaria and inflammatory diseases such as arthritis or lupus. However, HCQ studies on SARS-CoV-2-infected patients were poorly designed, and sometimes, with a smaller number of patients. Moreover, if a study used a large number of patients, but lacked statistical/scientific measures or ethical issues, their outcome results were not reliable. This is what happened to the study published in The Lancet, which compelled the authors to retract their study in month of May 2020. 

#### 6.2.3. Type I IFN Receptors

Inhibition of IFN receptors may block the entire JAK/STAT signaling by preventing the binding of type I IFNs to their receptors. It has been shown that genetic ablation of type I IFNs or depletion of inflammatory monocytes-macrophages (IMMs) protects mice from lethal infection of SARS-CoV, without affecting the viral load. This group demonstrated that type I IFNs and IMMs promote viral infection and identified potential therapeutic targets for including SARS-CoV-2 and other respiratory viruses [21]. To block type I IFN signaling, IMMs, and M1 polarized macrophages, the best option would be preventing the binding of IFNs at their receptors by blocking them. Anifrolumab, a monoclonal antibody of IFNα receptor 1 (IFNAR1), is used to treat inflammatory autoimmune disease systemic lupus erythematosus (SLE), in which the immune system attacks its own tissues by blocking the pro-inflammatory type I IFNs pathway.

Anifrolumab antibody not only blocks IFNAR1, but also blocks type I IFN-dependent STAT1/STAT2 phosphorylation, and therefore, it suppresses type I IFN production by blocking the type I IFN auto-amplification loop, and subsequently, it inhibits the proinflammatory cytokine ‘storm’ [125].

#### 6.2.4. STAT1/2

Type I IFNα/β receptors (IFNAR) bind to ligands IFNs in an autocrine or paracrine manner, which led to the activation of intracellular phosphorylases Janus Kinases (JAK1) and TYK2. As discussed in the previous section, JAK1 and TYK2 phosphorylate STAT1 and STAT2. Together with IRF9, it phosphorylates STAT1 and STAT2 and forms a homo- or heterotrimeric ISGF3 complex. In the nucleus, the ISGF3 complex binds ISRE containing the promoter of ISGs including ACE2. These ISGs have diverse cellular functions to promote the antiviral state in the cell. Inhibitors of STAT1 or STAT2 prevent the nuclear translocation of the complex [126]. Inhibition of STAT1 could potentially be a new molecular target of an anti-inflammatory treatment by preventing the interaction to STAT2. Inhibition of specific kinases is required that phosphorylate STAT1 and STAT2 at tyrosine residue and make them enzymatically active to form the ISGF3 complex. A combination of STAT1/2 inhibitors may also prevent ISGF3 formation eventually to prevent the cytokine ‘storm’ and expression of ACE2 expression.

#### 6.2.5. JAK1/TYK2

The JAK/STAT pathway is a major component of innate immune response in which JAK1 and TYK2 phosphorylate STAT1 and STAT2.

JAK1 is important for the IL6 and type-I interferon (IFN) family, while TYK2, in addition to type-I IFN signaling, also plays a role in IL23 and IL12 signaling. JAK family members are tyrosine kinases, namely JAK1, JAK2, JAK3, and TYK2. These kinases get activated through conformational changes due to ligands (IFNα/β) and IFAR1/R2 receptors binding, which initiates the phosphorylation cascade and leads to the activation of STATs proteins. Therefore, JAK family kinases are potential targets, and inhibiting their functions could prevent the signaling pathway and expression of ACE2. Specific JAK inhibitors, such as Filgotinib, Upadacitinib, and PF04965842, or TYK2 inhibitors and BMS-986165, are currently in clinical development for autoimmune diseases rheumatoid arthritis (RA), psoriasis, and Crohn’s disease [127]. Monoclonal antibodies have superior specificity than inhibitors and demonstrated better efficacy in Phase III clinical trials of psoriasis, psoriatic arthritis, inflammatory bowel disease, and rheumatoid arthritis studies, leading to multiple drug approvals [128].

#### 6.2.6. IRFs

IRF family members are key mediators of the JAK/STAT signaling pathway and play a major role in the induction of cytokines of pro-inflammatory host cell’s immune response. There are five functional subgroups of IRFs, i.e., IRF1 and 2, IRF3 and 7, IRF4 and 8, IRF5 and 6, and IRF9 as a part of the ISGF3 complex [129,130]. In the activation of type I ISGs transcription during viral infection, IRF1, -3, -5, and -7 play a pivotal role [131]. In contrast, IRF2 and IRF4 has been reported to suppress type I IFN signaling [132]. IRF functions can be interrupted at several levels in the cell to prevent the induction of cytokines and exploit it for the therapeutic purpose [133]. IRFs can be modulated directly or indirectly to perturb their functions. Strategies to direct IRF modulation include suppressing the expression at transcription or post-transcription or protein level by inhibition of intermediate components involved in the expression (activators or regulators) or inhibition of the protein itself enzymatically by synthetic or natural compounds. Mechanisms by which these compounds modulate the IRF activities are blocking the ligand binding to TLR or IFN receptors and preventing the phosphorylation of STAT1/STAT2 for downstream signaling. Furthermore, downstream of the JAK/STAT pathway, inhibition of ISGF3 complex formation or IRFs interaction with other transcription factors occurs. One potential clinical strategy could be by blocking IRF DNA binding sites on the promoter of ISGs [134].

#### 6.2.7. NFκB

Transcription factor NFκB is considered as a major factor in the proinflammatory signaling pathway, based on the role in the TLR-dependent innate immune response by controlling the expression of IFNs, subsequently proinflammatory cytokines, such as iNOS, interleukin 1 (IL1), IL12, IL27, and tumor necrosis factor α (TNFα) [135]. NFκB activation induces various target genes, such as pro-proliferative, cytokines (IL6), type I IFNs, and anti-apoptotic genes, and NFκB signaling crosstalk affects major cellular functions controlling signaling pathways, including those involving STAT1, STAT2, STAT3, AP1, IRFs, NRF2, Notch, WNT–β-catenin, and p53 [136]. p53 and NFκB co-regulate the pro-inflammatory gene (such as IL6, IL12, and IL27) responses in human macrophages [137]. NFκB, together with PARP1, is responsible for the activation of the iNOS and P-Selectin upon stimulation of the cells with inflammatory stimuli in vivo [138,139]. PARP1 functions synergistically with NFκB cofactors (RelA/p65, RelB, c-Rel, p50 (NFκB1), and p52 (NFκB2)) in stabilizing the interactions between NFκB and the transcription initiation complex, thereby facilitating the formation and activation of transcription in vivo. Therefore, NFκB is a central regulator of innate immune response and, together with PARP1 enzymatic activity, plays a major role in NFκB-dependent gene [140].

Without going into detail into the functions of NFκB, we emphasize here its inhibitors. Inhibition of NFκB signaling is not a new approach. However, the number of reported inhibitors of NFκB signaling are staggering. They are broadly divided according to their functions in cells, targeting NFκB signaling at the following levels: upstream of IKK (at a receptor or adaptor level), IKK complex or IκB phosphorylation, cytoplasmic ubiquitination or proteasomal degradation of IκB, translocation of NFκB in the nucleus, NFκB DNA binding, and NFκB-directed gene transactivation [141,142].

A potential unique approach to NFκB inhibition by RNA interference (RNAi) to suppress the expression of subunits of NFκB and production of it’s dependent proteins, and their cellular functions eventually. Other similar approaches, including small interfering RNA (siRNA), small hairpin RNA (shRNA) and morpholinos, all approaches silence the genes post-transcriptionally, thereby disrupting the functions of target proteins effectively. Furthermore, antisense DNAs effectively block the expression of NFκB target genes. Although inhibition of NFκB in various manners to treat inflammatory diseases is a promising therapeutic strategy, to use these inhibitors on SARS-CoV-2 patients, there should be multiple factors to take into account, since NFκB is a central regulator of many biological functions of cells other than the pro-inflammatory response, including apoptosis, DNA damage repair, and cellular growth.

#### 6.2.8. ACE2

There are several steps in the virus life cycle from viral entry into host cells, replication of viral genome, and assembly of capsid and viral genome, which are potential targets for therapeutics [143]. ACE2 inhibition can be achieved by blocking either enzymatic functions by ACE2 inhibitors or transcription suppression by disrupting the upstream type I IFN pathway or JAK/STAT pathway by inhibiting STAT1/STAT2, JAK1/TYK2, or IRFs, as discussed before. ACE2 receptors are present in epithelial cells and act as a protective line for primary barriers in lungs, blood vessels, gastrointestinal tract, heart, kidney, and liver for microbial infections. 

It is a good strategy to inhibit implantation of SARS-CoV-2 on host cell by decreasing the expression, release, and/or activity of ACE2 or TMPRSS2. Coronaviruses (SARS-CoV, SARS-S, and SARS-CoV-2) comprise two surface glycoproteins, i.e., S1 and S2. The S1 glycoprotein contains a receptor-binding domain (RBD) and S2 contains a fusion peptide. These viruses enter the host cells through interaction between RBD and ACE2 receptors [144], followed by endolysins in endosomes at low pH in cathepsin L, which cleaves S1 receptors and exposed the S2 domain of spike protein for membrane fusion [145,146]. However, this fusion can take place independently from pH, as reported by a previous study [147]. Several synthetic inhibitors have been identified that target surface receptors of viruses from previous studies of SARS-CoV from 2003 to now. One of the ACE2 inhibitor Ramipril is in a phase 2 clinical trial to examine the efficacy in reducing the mortality of patients (Ramipril is a prodrug that converts in liver cells to its active metabolite Ramiprilat). 

#### 6.2.9. TMPRSS2

TMPRSS2 inhibition can be achieved by either inhibition of its enzymatic activity to prevent priming of S protein or to repress its expression by transcription regulation, which we discussed below. TMPRSS2 inhibition is a safe and efficient treatment of viral infections (MERS-CoV, SARS-CoV, SARS-CoV-2, several H1N1 subtypes of influenza A viruses, and Asian H7N9 influenza virus) that utilize TMPRSS2 for implantation in host cells [148,149,150]. Bromhexine hydrochloride is shown to inhibit protease activity of TMPRSS2 in metastatic prostate cancer of mice, and accordingly, it could also be used for the treatment of both influenza and coronavirus infections [150]. Aprotinin, the basic trypsin inhibitor of the bovine pancreas, is a 58 amino acid broad-spectrum serine protease inhibitor [151]. Camostat, an oral serine protease inhibitor, is used for the treatment of postoperative reflux esophagitis and chronic pancreatitis [152]. TMPRSS2 inhibitors are in clinical trials to evaluate the possible therapeutic effects of Camostat, Nafamostat, and Bromhexine on the course and outcomes of COVID-19.

#### 6.2.10. Viral Genome

##### Antisense

Antisense RNA technology is an emerging area in clinical research, several groups are using this technology in their diseases research programs (cancers, myopathies, and huntington disease) [153]. Antisense RNA could be a possible therapy for SARS-CoV-2 as an antiviral drug by binding and blocking the translation and cleaving of the viral genome [154]. This therapy can be developed by the use of 30 bases of modified oligonucleotides targeting virus mRNAs of envelope protein, nucleoprotein region, and RNA-Dependent RNA Polymerase (RDRP) [155,156]. These complementary modified primers for envelope or nucleoprotein region can be delivered by using liposome or another suitable vector to deliver lung epithelial cells to target viral mRNA of envelop or nucleoprotein and block viral multiplication using two possible mechanisms: (1)Complementary oligo hybridizes with virus mRNA, blocking the translation of the virus protein, finally stopping virus multiplication (Figure 5A) [154](2)Complementary oligo hybridizes with viral mRNA to form duplex, after that RNAH enzyme can be used to degrade the duplex. Partially degraded mRNA will be unable to translate in to viral protein; therefore, virus cannot propagate.

Newly developed, ribozyme [157] or riboswitch [158] can also be used to block the translation of the virus. Ribozyme is a 50 to 100 bp long nucleotide that binds and cleaves after the target binding. Ribozyme and riboswitch can be used to prevent nonspecific binding of the target. Small complementary size ribozyme or riboswitch against viral envelope protein [159], viral nucleoprotein or 3′ of the viral RNA make them to express easily in adenoviruses, a potential possibility to develop vaccine. Synthetically engineered complementary RNA of these genes could be express in the lentiviruses (Figure 5B) or liposome-mediated delivery of antisense RNA in to lung epithelial cells, can be used (Figure 5C). Viral infected host cells will express complementary ssRNA, possibly block the translation of SARS-CoV-2 envelop or nucleoprotein, and will finally slow down the viral replication or target 3′-5′ of the viral genome to block virus RNA replication (Figure 5B).

##### Bacteriophage-Based Vaccine

A bacteriophage is a virus that often infects and replicates using bacterial cells. Bacteriophages can be used as a tool to deliver anti-sense DNA to viruses as DNA vaccines and are safer to host cells. 

Vaccines can be designed either for DNA engineering in phage-displayed or phage genomic DNA. Phage display methods could be used to develop a new vaccine for SARS-CoV-2 [160,161]. In this method, a specific protein or peptide displays on the surface of the phage and delivers the immunogenic peptides to the target cells. Some phages are already in use for phage display technology, such as T7, M13, and lambda display system [160]. Antigenic protein or peptide cistronically fuse with phage coat protein and express with coat protein. The recombinant phage can be engineered by inserting DNA with the phage coat protein during the process of infection in eukaryotic cells. The antibody produced by the phage display could be humanized by producing in Chinese Hamster Ovary (CHO) cells and used them as a specific target to block the binding of spike protein with ACE2.

A vaccine approach can be designed similar to the adenovirus dependent UK’s Oxford University and AstraZeneca. The viral RNA of the spike protein of SARS-CoV-2 is integrated into the genome of adenoviruses. The adenovirus acts as a carrier or vehicle to infect host cells. Once entering in to host cells, viral RNA uses the host cell’s ribosome and translation machinery to synthesize spike proteins of SARS-CoV-2. These spike proteins internalize by antigen-presenting cells (APCs) and express on the outside of the cells and recognized by MHCI and MHCII and initiate the acquired immunity response in host cells for future, real SARS-CoV-2 infection.

##### Phage DNA Vaccine

A phage DNA vaccine can be delivered in SARS-CoV-2 target cells (lungs cells). Viral spike protein can be engineered in the phage genome with the eukaryotic promoter and expression systems (157). The engineered phage can be delivered to the target cells and express an antigenic protein that can elicit an immune response for the prevention of SARS-CoV-2.

##### CRISPR

CRISPR could be an effective tool and can be used for the antiviral activity of SARS-CoV-2. Its positive sense ssRNA that can be targeted in conserve regions by CRISPR guide RNA (crRNA). This guide RNA can attract CRISPR/Cas9 to bind and degrade the virus genome. This approach can reduce virus propagation in host cells. Viral spike (S) protein, envelope (E) and nucleocapsid region, mRNA of RDRP, and nucleoprotein are included in potential targets list [162].

## 7. Discussion

This review provides evidence of the transcriptional regulation of SARS-CoV-2 receptor ACE2 and TMPRSS2 genes. The expression of these genes determines the entry of viruses into host cells. It opens a new avenue to identify new targets to block their expression, and eventually block the host cell entry of SARS-CoV-2. However, the ACE2 receptor can be blocked by a clinically proven inhibitor of ACE or TMPRSS2. Here, we discuss, based on published reports, PARP1-dependent TMPRSS2 regulation through auto-poly(ADP-ribosyl)ation, since viral infection stimulates pro-inflammatory pathway via the induction of iNOS and cytokines (IL6), increases the ROS in cells, and causes ssDNA damage [163,164]. Through enzymatic activation of PARP1, DNA damage repair machinery recruits to the damaged site. Auto-poly(ADP-ribosyl)ated PARP1 moves away from the binding site of the promoter of TMPRSS2. PARP1 not only regulates transcription of TMPRSS2, also regulates type I IFN pathway genes, such as IFNβ, STAT1 and STAT2 by binding either the upstream or downstream region [37]. Using non-NAD^+^ PARP1 inhibitors [46] could be important to block the virus entry in the cell by disrupting viral S protein priming and the IFN pathway.

Binding partner proteins of ISGF3 (STAT1, STAT2, and IRF9) are also part of the initiation of the transcription of ACE2. Therefore, targeting the expression of these proteins should be considered in the development of new therapies. In addition, NFκB is a major regulatory protein involved in various pathways, including transcription control of IFN genes (Figure 6). Although, inhibitors of NFκB subunits are already in use to control inflammation or cancer progression, new studies are warranted to look at their effects on virus-mediated infection. Boosting the immunity by vaccination is not a new strategy; however, in the current pandemic, the immune response of South-East Asian countries is much better than other countries, where BCG vaccination is not in practice. One of the most striking observations of this review is summarized in Figure 4A,B. Codon 72 polymorphism of p53 simply divide the world by their proximity to the equator, with mortality rates several folds higher in those countries (European and North American) with an R72 population (further north to the equator) compared to countries (Asian, African, South American, and Australia) with a P72 population. This observation needs to be confirmed by the SNP of p53 in the patients of these countries. If this is true, then the selection of the P72 variant near the equator during evolution is justifiable, as these countries are more vulnerable to various viral or bacterial pathogenic infections. 

Finally, the antisense and CRISPR technology to target viral RNA or RDRP are very promising, although these approaches are in very early stages. Other new approaches could be potential novel treatments, such as inhibitors targeting the ACE2 receptor or TMPRSS2, IFN pathway genes or NFκB targeted drugs, based on codon 72 polymorphism of p53 or because these proteins are integral part of internalization process of virus in to host cells. But these approaches remain promising hypotheses until they are scrutinized under nonrandomized rigorous testing for efficacy and safety by randomized controlled trials in humans.

The pathogenesis of SARS-CoV-2 infection not only affects the lungs, it also causes neurological disorders; we will learn about its long-term pathogenic effects in future. With due course of time, we will gradually understand better the complex immune response by host cells. Currently, treatment strategies based on repurposing the known drugs which were approved for other diseases [154], previous or under study, include mRNA anti-sense, receptor inhibitors, immunomodulators, specific antibodies, and adjunctive therapies. Anti-inflammatory properties of dexamethasone brought it to the first line of successfully tested drugs in a study in the United Kingdom (UK) and are used because it improved survival in COVID-19 patients [165]. Preliminary results of this trial show dexamethasone reduced deaths by one-third in ventilated patients [122]. In the USA and Europe, mRNA-based vaccines of Moderna Therapeutics and Pfizer-BioNtech are in clinical use. The Moderna’s mRNA-1273 vaccine showed 94.1% efficacy at preventing COVID-19 illness, including severe disease [166]. Pfizer-BioNtech’s two doses of BNT162b2 mRNA vaccine presented 95% protection against COVID-19 in persons of 16 years of age or older [167]. UK’s Oxford University and AstraZeneca’s vaccine shows the prompted immune response to neutralize the virus and is in clinical use [168]. As the third vaccine in the United States, the FDA, on 27 February 2021, issued emergency authorization for a COVID-19 vaccine developed by Johnson & Johnson. This vaccine reduces the cases of moderate to severe COVID-19 infection by 66.1%, starting 28 days after the single shot. Results from a study from other countries suggest that the vaccine worked better in some regions of the world than others. In the U.S., the vaccine was 72% protective. As of now, vaccination of American individuals 18 years old and up (at least one dose) has reached 69.0% with all three available vaccines and 60% people are fully vaccinated (Coronavirus Resource Center, John Hopkins University). 

Despite vaccines being in development and in clinical use, social practices will be relevant in the next few years and a clinical exploration to develop novel treatments should be continued for multiple new variants of SARS-CoV-19 that have emerged or emerging, such as the variant identified in the UK, called B.1.1.7 (Alpha), the variant identified in South Africa called B.1.351 (Beta), which emerged independently from B.1.1.7 in Brazil, which is a variant called P.1 (Gamma), the variant B.1.617.2 lineage, first identified in India, which is now called Delta, Delta plus (some places in USA) and the Midwest variant (S Q677H) [169,170,171,172]. In the beginning of 2021, new cases have dropped significantly in the entire nation as people started to get vaccines. Now, the USA from east to west coast is slowly returning to a new normal; however, the recent spike of infection rate of the Delta variant of SARS-CoV-2 in unvaccinated population, it increases the fear of a new wave of infections in people again, and experts started to call it pandemic of unvaccinated people. If it continues, then we will return to where we were last year because of the hesitancy of people to get vaccinated.

As we discussed, inhibition of ACE2 through a pro-inflammatory pathway or PARP1 inhibitors to target TMPRSS2 could be a potential strategy to decrease viral entry and reduce the mortality rate. 

In summary, this review provides key insights into the transcriptional regulation of ACE2, the putative role of PARP1-dependent expression of TMPRSS2, IFNβ, and STAT1/2, and the innate immune response. We emphasizes for a comprehensive research study, should be conducted on the mortality rates or immunogenic response discrepancy based on the geographical distribution of the P72 or R72 population to demonstrate further. We anticipate that inhibition of ACE2 through a pro-inflammatory pathway or PARP1 inhibitors to target TMPRSS2 could be a potential strategy to decrease SARS-CoV-2 entry and reduce the mortality rate. The review highlights potential targets or therapies for antiviral intervention.

## Figures and Tables

**Figure 2 ijms-22-08660-f002:**
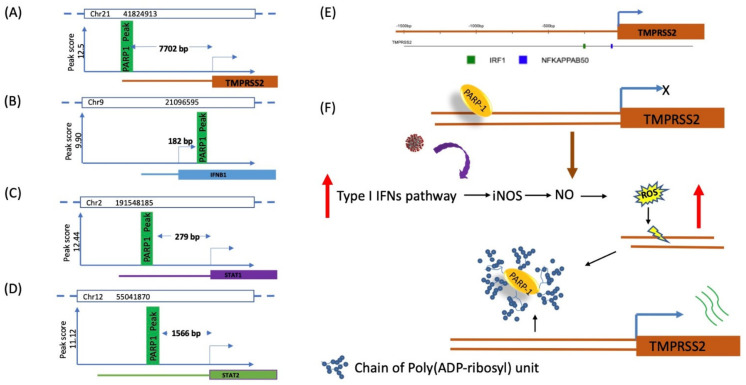
Binding sites of PARP1 on promoter or downstream regions of different immune-regulatory genes. PARP1 binds to TMPRSS2, IFNβ, and JAK/STAT pathway genes either upstream or downstream to TSS. These PARP1 binding sites were reported by Lodhi et al. 2014 [37] and derived from their ChIP-seq data. Chromosome numbers and PARP1 binding locations are indicated in each figure. (**A**). PARP1 binds to about 7.7 kb upstream to TSS of TMPRSS2. (**B**). PARP1 binds just 182 bp downstream to TSS IFNβ promoter. (**C**). PARP1 binding to STAT1 promoter in the upstream region at 279 bp; it is not known how PARP1 is involved in the transcription regulation of this gene. (**D**). There is a PARP1 binding site in the upstream region around 1.5 kb promoter of STAT2. (**E**). Using www.interferome.org, we generated transcription binding sites in the promoter region of TMPRSS2, which has been developed using interferome (www.interferome.org). The location of transcription factors on each of the genes from the region spanning −1500 bp to +500 bp from the start site. Each coloured box represents a specific transcription factor, the key for which is provided below the graphic. TRANSFAC* predicted a match between the transcription factor and its predicted binding [34]. (**F**). Induced enzymatic activity of PARP1 auto-poly(ADP-ribosyl)ates, subsequently followed by SARS-CoV-2 infection. Virus internalization through ACE2 receptors and priming of viral S protein by residual TMPRSS2. TLR3/7 recognizes the PAMPs and induces the type I IFN pathway, which leads to the production of iNOS and, therefore, NO. Increase of NO creates ROS in cells, which causes DNA damage and a reason for PARP1 activation. Auto-poly(ADP-ribosyl)ated PARP1 moves away from the binding site, and it may increase the transcription of TMPRSS2, hence facilitating virus entry.

**Figure 3 ijms-22-08660-f003:**
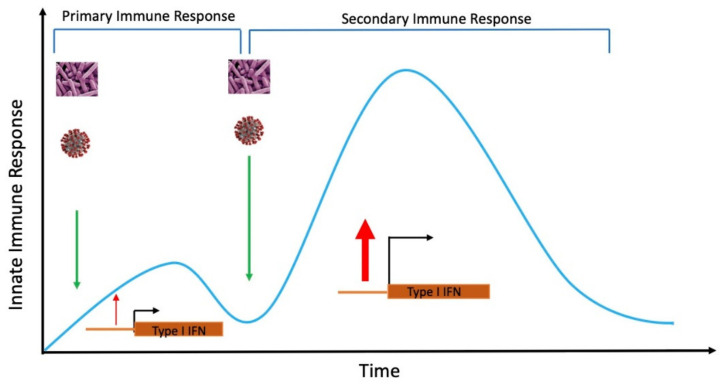
Vaccination to train host immunity improves the anti-viral or bacterial host defense by enhancing the innate immune response. Type I IFN response enhances primary infection, which is further enhanced by several folds after secondary infection.

**Figure 4 ijms-22-08660-f004:**
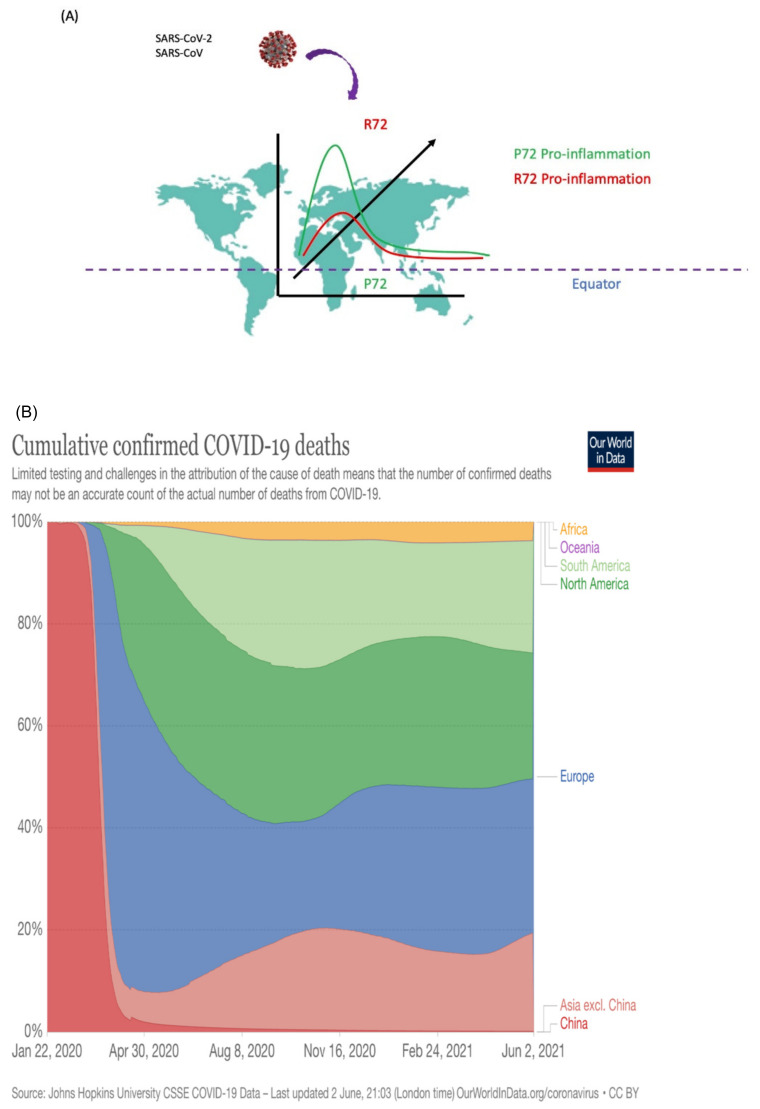
(**A**). Model for the relationship between the people with codon 72 SNP of p53 variants of earth’s northern latitude or equator region and inflammatory response to viral (SARS-CoV, SARS-CoV-2)/bacterial infections. (**B**). Relation between the confirmed mortality rate of virus-infected patients and proximity to the equator. It aligns well with the innate immune response and geographical distribution of codon 72 polymorphism of p53. The mortality rate is much higher in countries further north to the equator, with R72 variant populations, compared to countries near the equator, with P72 variant populations.

**Figure 5 ijms-22-08660-f005:**
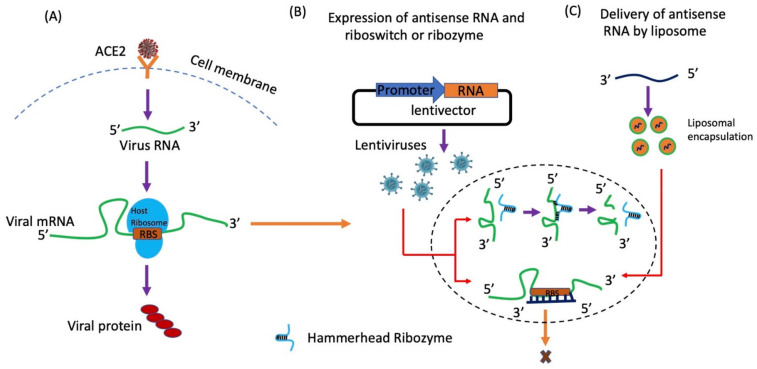
Antisense RNA approach to block translation of SARS-CoV-2′s mRNA. (**A**). Entry of SARS-CoV-2 inside the cell and virus mRNA released and mRNA of virus translate the virus protein. (**B**). The lentivirus expressing antisense RNA or riboswitch or ribozyme that can bind on virus mRNA and block translation. (**C**). Delivery of antisense RNA by liposomes. Expression of antisense RNA can block the translation of viral protein.

**Figure 6 ijms-22-08660-f006:**
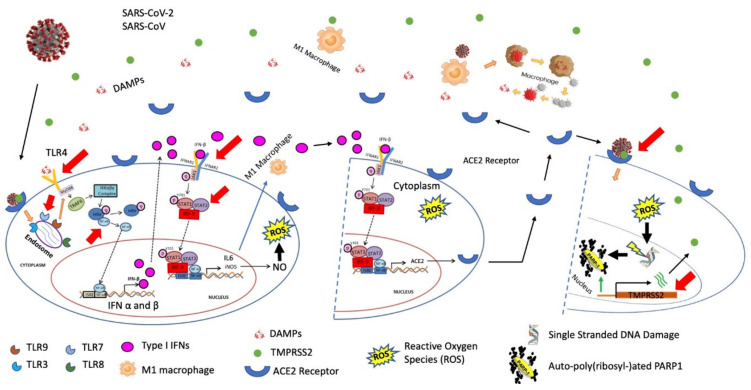
Overview of PAMPs/TLR9, type I IFN, and JAK/STAT signaling mediate the activation of pro-inflammatory cytokines and ACE2, and activation of PARP1 by sensing DNA damage from the induction of ROS to facilitating the transcription of TMPRSS2. ACE2 receptors on the surface of host cells recognize S1 glycoprotein, which contains receptor binding domain (RBD), and TMPRSS2 primes S2, containing fusion peptide. It can achieve entry via the plasma membrane or endosome. The presence of exogeneous and membrane-bound proteases, such as trypsin and TMPRSS2, triggers the early fusion pathway (not shown). In parallel, host cells endocytosed it by cleaving at the S1/S2 site. After entry into the lung’s alveolar epithelium, endosomal single-stranded (ss)RNA sensors (TLR7 and TLR8) sense the viral genomes and recruit the adaptor proteins, MyD88, for downstream signaling. MyD88 activates TRAF6 as well as the transcription factor NFκB and induce expression of IFN-β cytokine. Through autocrine, Type I interferon (IFNβ) binds to the IFN-α/β receptors to phosphorylate and activate JAK1. This kinase further phosphorylates STAT1 (Y701) and STAT2 (Y). JAK/STAT signaling initiates through the phosphorylation of STAT1 and STAT2, a physical interaction with IRFs to form the ISGF3 complex (STAT1, STAT2, and IRF9) and regulates the expression of ISGs. ISGF3 enters the nucleus and directly binds to the upstream regulatory regions (IFNs) of IFN-sensitive genes (iNOS) and cytokine (IL6) and causes a cytokine ‘storm’, thereby regulating the expression of specific gene expressions of M1 phenotype, which leads to M1-polarized macrophages. DAMPs bind to TLR4 receptors after phagocytosis of bacteria by M1 polarized macrophages. Simultaneously, type I IFNs binds to neighboring cells by paracrine signaling and via the similar JAK/STAT pathway, while the ISGF3 complex binds to ISRE of the ACE2 promoter to initiate the expression. Production of NO increases the ROS in the cell, which leads to ssDNA damage. It enzymatically activates auto-poly(ribosyl)ation of PARP1 and accumulation of negatively charged pADPr repulse poly(ribosyl)ated PARP1 from DNAs (TMPRSS2, IFNβ and STAT1/2), mediating loosening of chromatin and facilitating the transcription initiation of these genes. Potential targets ACE2, TMPRSS2, NFκB, STAT1/2, IRF9 (ISGF3 complex), TLR9, and type I IFN receptors are indicated by a red arrow. Nucleotide sequences in promoter of ISGs are termed as interferon-stimulated response elements (ISREs).

## Data Availability

Supporting information about CoVID19 can be obtained from https://ourworldindata.org, https://covid19.who.int, https://coronavirus.jhu.edu/map.html and https://www.ecdc.europa.eu/en, and transcription binding sites and Gene Ontology (GO) analysis of ACE2 can be obtained from http://www.interferome.org/interferome/home.jspx.

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
