# Peer review of "SARS-CoV-2: Understanding the Transcriptional Regulation of ACE2 and TMPRSS2 and the Role of Single Nucleotide Polymorphism (SNP) at Codon 72 of p53 in the Innate Immune Response against Virus Infection"

_ijms, 2021, doi:10.3390/ijms22168660_

Round 1
Reviewer 1 Report
The present review is a good entry that explores the regulation of ACE2 and TMPRSS2 at transcriptional level. The main input relies on polymorfism analysis at codon 72 of p53 in innate immune response to counteract SARS-CoV2 infection. Figures are informative.
The manuscript is well organized in sections that have been properly discussed.
Indeed the introduction is too long and less focused on the expected aim. I suggest a careful revision; remove lines 98-107, as an example, since the same information may be derived from previous sentences (see lines 31-50).
Lines 69-71 related to bacteria could be removed, leaving the attention only to SARSCoV2.
Modify SARSCoV-19 into COVID-19 and SARSSCoV2, respectively, in line 115 and 766.
Please include to citations 55-57 that of prion protein PrP90-231 causing the overexpression of iNOS and ROS. REF: V. Villa et al. / Pharmacological Research 113 (2016) 500–514.
Often English style is difficult to read; please revise all the paper main text!
Some examples of errors are as follows:
line 146: modify compare to with compared to
line 157: modify in "none of the factors"
line 294: modify in " R72 variant is metabolically tolerated in the northern..."
line 310: remove "this polymorphism shows"
line 311: modify "there is" in "where"
line 730: modify in "needs to be confirmed"
line 172: remove "in" after binding
I consider the paper of interest after addressing the indicated revisions
Author Response
Dear Reviewer,
We thank you eviewer for his/her time to view our manuscript. We carefully went through the comments and suggestions to include in revised manuscript. We hope our article reach wide global readers.
Please find point wise response to your comments.
The present review is a good entry that explores the regulation of ACE2 and TMPRSS2 at transcriptional level. The main input relies on polymorfism analysis at codon 72 of p53 in innate immune response to counteract SARS-CoV2 infection. Figures are informative.
The manuscript is well organized in sections that have been properly discussed.
Response: We are very thankful to reviewer for his/her constructive comments and suggestions in the entire review process. We hope that our revision satisfies the reviewer and matches the standard to publish in IJMS journal.
Indeed, the introduction is too long and less focused on the expected aim. I suggest a careful revision; remove lines 98-107, as an example, since the same information may be derived from previous sentences (see lines 31-50).
Response: We agree with reviewer and thank him/her for suggestion. We revised carefully sentence by sentence and removed the overlapping information as suggested by you. It will certainly improve quality of our manuscript.
Lines 69-71 related to bacteria could be removed, leaving the attention only to SARSCoV2.
Response: We thank him/her for suggestion, however, we respectfully are not fully agree with reviewer. To understand better for readers, we described both cell surface and intracellular TLRs, cell surface TLRs recognize lipoproteins, lipids and proteins of bacterial or microbial membranes whereas intracellular TLRs recognize nucleic acids derived from viral infections. We understand reviewer's concern to focus only on viral infection part but we think readers should know about TLRs responsible for bacterial infection and understand the difference. In figure 6 we mentioned both type of TLRs and their role in bacterial or viral infection pathway.
Modify SARSCoV-19 into COVID-19 and SARSSCoV2, respectively, in line 115 and 766.
Response: We thank reviewer to attract our attention on important part of review. We corrected all the SARS-CoV-2 in to CoVID19 as suggested.
Please include to citations 55-57 that of prion protein PrP90-231 causing the overexpression of iNOS and ROS. REF: V. Villa et al. / Pharmacological Research 113 (2016) 500–514.
Response: In revised version of manuscript we included all the suggestion of reviewer and recent study and added (Ref. 58).
Often English style is difficult to read; please revise all the paper main text!
Response: We agree with reviewer and thank him/her for suggestion. We have reviewed every sentence in the manuscript and revised the wordings carefully. Certainly, it will improve quality of our manuscript.
Some examples of errors are as follows:
line 146: modify compare to with compared to
line 157: modify in "none of the factors"
line 294: modify in " R72 variant is metabolically tolerated in the northern..."
line 310: remove "this polymorphism shows"
line 311: modify "there is" in "where"
line 730: modify in "needs to be confirmed"
line 172: remove "in" after binding
Response: In revised version of manuscript, we included all the suggestion of reviewers and it will certainly improve the quality of our manuscript.
I consider the paper of interest after addressing the indicated revisions
Response: we are very thankful to reviewer. His/her constructive comments and suggestions certainly will improve the quality of manuscript and will make easy to understand to readers. We hope that our revision satisfies the reviewer and matches the standard to publish in IJMS.
Thank you and respectfully submitted,
Niraj
Reviewer 2 Report
Lodhi et al presented an interesting review entitled "SARS-CoV-2: Understanding the Transcriptional Regulation of ACE2 and TMPRSS2, and the Role of Single Nucleotide Polymorphism (SNP) at Codon 72 of p53 in Innate Immune Response Against Virus Infection". The authors discussed the following points: a) transcriptional regulation of ACE2 by ISGF3 assembly complex in type I interferon and JAK/STAT pathway: b) transcriptional regulation of TMPRSS2 by PARP1 binding. c) Modulation of immune response to Virus, including immune response training to develop the immunity and immunity development during virus evolution. d) modulation of immune response genes to develop therapy. e) prevention and new targets to develop therapy including vaccination, effect of codon 72 polymorphism of p53 , and potential new targets such as PARP1, TLR 3, 4 and TLR 9, type I IFN Receptors, STAT1/2, JAK1/TYK2 , IRFs, NFkB, ACE2 , TMPRSS2 b, and viral genome targets.
Overall the review is comprehensive, well designed, and has an acceptable flow.
Minor suggestions:
The review includes many abbreviations. I would suggest the authors to include list of abbreviation in the beginning of the review to help the readers to understand the digest the review well.
Author Response
Dear Reviewer,
We thank you eviewer for his/her time to view our manuscript. We carefully went through the comments and suggestions to include in revised manuscript. We hope our article reach wide global readers.
Please find point wise response to your comments.
Lodhi et al presented an interesting review entitled "SARS-CoV-2: Understanding the Transcriptional Regulation of ACE2 and TMPRSS2, and the Role of Single Nucleotide Polymorphism (SNP) at Codon 72 of p53 in Innate Immune Response Against Virus Infection". The authors discussed the following points: a) transcriptional regulation of ACE2 by ISGF3 assembly complex in type I interferon and JAK/STAT pathway: b) transcriptional regulation of TMPRSS2 by PARP1 binding. c) Modulation of immune response to Virus, including immune response training to develop the immunity and immunity development during virus evolution. d) modulation of immune response genes to develop therapy. e) prevention and new targets to develop therapy including vaccination, effect of codon 72 polymorphism of p53 , and potential new targets such as PARP1, TLR 3, 4 and TLR 9, type I IFN Receptors, STAT1/2, JAK1/TYK2 , IRFs, NFkB, ACE2 , TMPRSS2 b, and viral genome targets.
Overall the review is comprehensive, well designed, and has an acceptable flow.
Response: We are very thankful to reviewer for his/her constructive comments and suggestions in the entire review process. We hope that our revision satisfies the reviewer and matches the standard to publish in IJMS journal.
Minor suggestions:
The review includes many abbreviations. I would suggest the authors to include list of abbreviation in the beginning of the review to help the readers to understand the digest the review well.
Response: In revised version of manuscript we included abbreviation list in the beginning as suggested by reviewer and we are very thankful to reviewer. His/her constructive comments and suggestions certainly will improve the quality of manuscript and will make easy to understand to wide global readers.
Thank you and respectfully submitted,
Niraj